# HyperNetwork-based Decoupling to Improve Model Generalization for Few-Shot Relation Extraction

**Liang Zhang**[1,2*]  **Chulun Zhou**[3*]  **Fandong Meng**[3]  **Jinsong Su**[1,2†]
**Yidong Chen**[1,2†] and **Jie Zhou**[3]

[1]School of Informatics, Xiamen University, China
[2]Key Laboratory of Digital Protection and Intelligent Processing of Intangible Cultural Heritage of Fujian and Taiwan (Xiamen University), Ministry of Culture and Tourism, China
[3]Pattern Recognition Center, WeChat AI, Tencent Inc, China
lzhang@stu.xmu.edu.cn, chulunzhou@tencent.com, {jssu,ydchen}@xmu.edu.cn

## Abstract

Few-shot relation extraction (FSRE) aims to train a model that can deal with new relations using only a few labeled examples. Most existing studies employ Prototypical Networks for FSRE, which usually overfits the relation classes in the training set and cannot generalize well to unseen relations. By investigating the class separation of an FSRE model, we find that model upper layers are prone to learn relation-specific knowledge. Therefore, in this paper, we propose a HyperNetwork-based Decoupling approach to improve the generalization of FSRE models. Specifically, our model consists of an encoder, a network generator (for producing relation classifiers) and the generated-then-finetuned classifiers for every $N$-way-$K$-shot episode. Meanwhile, we design a two-step training strategy along with a class-agnostic aligner, by which the generated classifiers focus on acquiring relation-specific knowledge and the encoder is encouraged to learn more general relation knowledge. In this way, the roles of upper and lower layers in our FSRE model are explicitly decoupled, thus enhancing its generalizing capability during testing. Experiments on two public datasets demonstrate the effectiveness of our method. Our source code is available at https://github.com/DeepLearnXMU/FSRE-HDN.

## 1 Introduction

Relation extraction is a fundamental task in information extraction that aims to identify the semantic relations between two entities in sentences (Zeng et al., 2020; Han et al., 2020). Typically, conventional approaches are highly dependent on a large amount of labeled data and cannot deal with unseen relations well. Therefore, recent studies (Han et al., 2018; Gao et al., 2019c) turn to Few-Shot Relation Extraction (FSRE) that only requires a handful

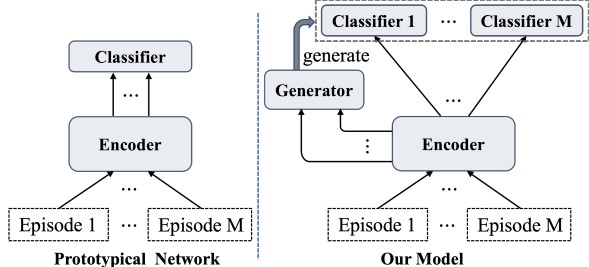

Figure 1: Illustration of the difference between conventional Prototypical Network model and our model.

of labeled instances and enhance the generalizing capability of FSRE model to new relations.

Current FSRE studies are usually conducted in an $N$-way-$K$-shot setting. In this setting, the model is trained on a series of episodes, each of which has $N$ relation classes. In each episode, every relation class includes $K$ support instances (as support set) and $Q$ query instances (as query set). Under this setting, most existing studies (Gao et al., 2019a; Yang et al., 2020; Zhang and Lu, 2022) employs Prototypical Network (Snell et al., 2017) for FSRE. The Prototypical Network is designed to learn a suitable prototypical vector for each relation using the support instances from the support set. Then, these vectors are used to predict the relation of the query instances. Moreover, to further improve the performance of the model, some recent studies (Han et al., 2021a; Liu et al., 2022) introduce relation description and adopt contrastive learning in Prototypical Network. Although these methods have achieved improvements, their models still tends to overfit relation classes appearing in the training set, and exhibit an unsatisfactory generalizing capability to unseen relations.

To further explore what restricts the generalizing capabilities of Prototypical Network models, inspired by Kornblith et al. (2021), we conduct a preliminary study to investigate the internal representations within different model layers. Specifically, we measure the layer-wise class separation

---

This work is done when Liang Zhang was interning at Pattern Recognition Center, WeChat AI, Tencent Inc, China.
*Equal contribution
†Corresponding author

between representations of instances belong into different relation classes during training. We observe that the representations present a higher class separation in the upper layers of the model than those in the lower layers. This intuitively shows that the upper layers are likely to learn relation-specific knowledge and overfit the relation classes in the training set, while the lower layers basically acquire more general relation knowledge. We speculate that this over-adaptation of upper layers limits the generalizing capability of the model.

Based on the above finding, in this paper, we propose a HyperNetwork-based Decoupling approach for FSRE. As illustrated in Figure 1, our model consists of three components: an encoder at the bottom, a network generator in the middle (for generating the initialized relation classifiers) and the generated relation classifiers at the top. In each episode, the network generator takes in the encoder output and generates an initialized relation classifier for current episode. Subsequently, the generated classifiers are fine-tuned, so that the model can quickly learn relation-specific knowledge and adapt to new relations. Meanwhile, the encoder and generator are encouraged to learn more general relation knowledge from the training set. In this way, we explicitly decouple the roles of the lower and upper components in our FSRE model.

To this end, our model update procedure consists of two steps: the fine-tuning of the classifier and the update of the encoder together with the generator. For the classifier fine-tuning, we first use the generator to produce an initialized classifier for every episode to discriminate the relation classes within them. Then, we use corresponding support set to optimize its generated classifier, which endows the classifier with relation-specific knowledge. Consequently, our model can effectively adapt to new relations through such produce-then-finetune process. Moreover, the update of the encoder and generator resorts to learning more general knowledge not specific to a certain episode, which is crucial for an FSRE model not to overfit. To better learn such knowledge, we train the model across different episodes so that it is not biased towards a narrow set of relations. Particularly, we treat a collection of $M$ sampled episodes as an updating interval for the encoder and generator. In each updating interval, the encoder and generator are jointly optimized through maximizing the overall performance of all trained classifiers on the query

set of these $M$ episodes.

During testing, when there is an episode containing unprecedented relation types not existing in the training set, we use the generator to produce a fresh classifier that is then fine-tuned by samples from the support set. Finally, only the encoder and the fine-tuned classifier are used to predict the relation of each query instance. By doing so, the model can quickly adapt to new episode.

However, as each updating interval contains only a handful of episodes, so that our model is still prone to bias towards relation classes in these episodes. To address this issue, we additionally design a *class-agnostic aligner*, which undergoes all episodes in the training set throughout the training process. Thus, with the help of the aligner, our encoder is able to learn more global general relation knowledge, further alleviating the overfitting to specific relations.

Experimental results on two public benchmarks show that our model consistently outperforms all competitive baselines. Extensive ablation and case studies demonstrate the effectiveness of various components in our model.

## 2 Preliminary Study

To better understand the limited generalizing capabilities of current FSRE models, inspired by (Kornblith et al., 2021), given a BERT-based Prototypical Network model, we first investigate layer-wise class separation between representations of instances belonging to different relations. Specifically, the class separation indicates the dispersion of representations belonging to the same class relative to the overall dispersion of all embeddings.

In each $N$-way-$K$-shot training episode, we measure the class separation in every layer of the model following the metrics in (Kornblith et al., 2021). Particularly, we denote the average within-class cosine distance and the overall average cosine distance in the $l$-th layer as $\overline{d}_{in}^l$ and $\overline{d}_{overall}^l$, respectively. They are calculated using the following equations:

$$\overline{d}_{in}^l = \sum_{n=1}^{N} \sum_{i=1}^{K} \sum_{j=1}^{K} \frac{1 - \text{sim}(\mathbf{h}_{n,i}^l, \mathbf{h}_{n,j}^l)}{NK^2}, \quad (1)$$

$$\overline{d}_{overall}^l = \sum_{n'=1}^{N} \sum_{n=1}^{N} \sum_{i=1}^{K} \sum_{j=1}^{K} \frac{1 - \text{sim}(\mathbf{h}_{n',i}^l, \mathbf{h}_{n,j}^l)}{N^2 K^2},$$

where $\mathbf{h}_{n,i}^l$ is the embedding of the $i$-th instance (from the $n$-th relation class) in the $l$-th layer, and

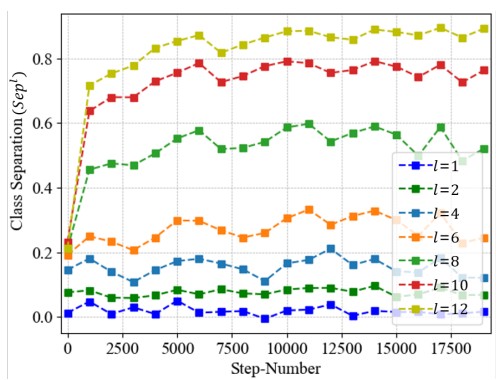

Figure 2: Layer-wise class separation $Sep^l$ in a 12-layer BERT-based Prototypical Network FSRE model with respect to the training steps.

$\text{sim}(\cdot, \cdot)$ is the cosine similarity. Then, $\overline{d}_{in}^l/\overline{d}_{overall}^l$ represents the relative within-class variance, a lower value of which corresponds to a higher degree of class separation in the $l$-th layer. Thus, the degree of class separation can be defined as $Sep^l = 1 - \overline{d}_{in}^l/\overline{d}_{overall}^l$.

As explored in Kornblith et al. (2021), higher class separation means the model tends to overfit the classes in the training set and is less transferable to unseen classes. Hence, we depict the layer-wise change of class separation with respect to the number of training steps in Figure 2. We can observe that $Sep^l$ in upper layers (*e.g.*, Layer-12 and Layer-10) quickly increase at early steps and consistently stay much higher than those in lower layers (*e.g.*, Layer-1 and Layer-2). Therefore, Figure 2 shows that the upper layers of the Prototypical Network model are likely to learn relation-specific knowledge and overfit the relation classes during training.

## 3 Methodology

In this section, we introduce our HyperNetwork-based Decoupling approach for FSRE. Specifically, we first provide the problem formulation of FSRE and give a detailed description of our model architecture. Then, we elaborate the two-step training strategy of our model. Moreover, we additionally design a class-agnostic aligner to further enhance the generalizing capability of our model. Finally, we describe how our model is used for test in an $N$-way-$K$-shot setting.

### 3.1 Problem Formulation

Current FSRE studies are usually conducted in an $N$-way-$K$-shot setting. In this setting, the model is trained and tested on a series of episodes, each of which is sampled from mutually exclusive training

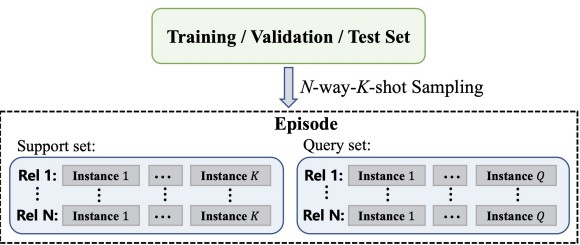

Context Sentence $x$: *Parias* was a son of *Philomelus*.

**Instance**: Relation Label $y$: ***Father***

Entities $e$: (*Parias*, *Philomelus*)

Figure 3: The illustration of sampling an $N$-way-$K$-shot episode from training/validation/test set.

and test sets, *i.e.*, relation classes for test do not exist in the training set. As shown in Figure 3, a sampled episode contains $N$ relation classes, each of which has $K$ instances (as support set $\mathcal{S}$) and $Q$ instances (as query set $\mathcal{Q}$). Meanwhile, each instance $(x, e, y)$ in the episode is comprised of a context sentence $x$, two entities $e=(e_h, e_t)$, and a relation label $y$, where $e_h$ and $e_t$ refer to head entity and tail entity, respectively.

### 3.2 Model Architecture

As shown in Figure 4, except the aligner, our model consists of three components: an encoder at the bottom, a network generator in the middle and the generated classifiers at the top.

**Encoder.** Following existing studies (Soares et al., 2019; Han et al., 2021a; Liu et al., 2022), we use BERT (Devlin et al., 2019) as the encoder $E$ of our model to encode both instances and relation descriptions. Specifically, for each instance, we first use four special tokens "[P1][/P1]" and "[P2][/P2]" to mark the start and end positions of the head and tail entities within an instance, respectively. Then, the encoder is employed to obtain a contextual representation for each token. Finally, we concatenate the representations of these tokens (i.e., [P1] and [P2]) at the start positions of the head and tail entities to form the representation of this instance: $\boldsymbol{h} = [\boldsymbol{h}_{\text{P1}}; \boldsymbol{h}_{\text{P2}}]$.

Moreover, as in (Liu et al., 2022; Li and Qian, 2022), we also encode relation descriptions and produce corresponding relation representations with the same dimensions as instance representations. Specifically, the representation of the relation description is obtained by concatenating the representation of the [CLS] token and the average representations of the remaining other tokens: $\boldsymbol{r} = [\boldsymbol{h}_{\text{CLS}}; \boldsymbol{h}_{\text{avg}}]$.

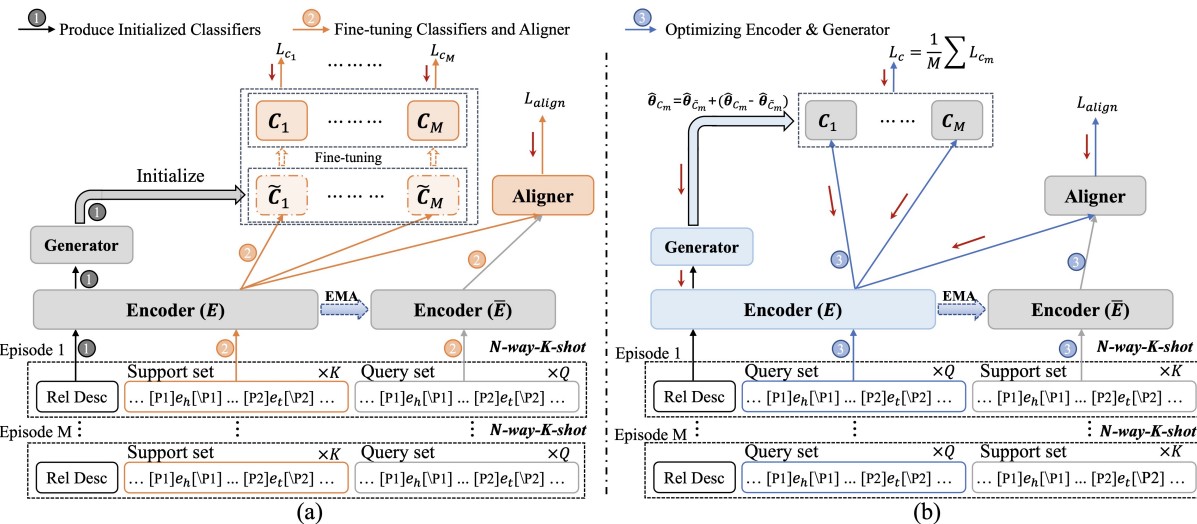

Figure 4: The architecture and two-step training framework of our model. In each episode, for the first step, the network generator of our model produces an initialized N-way relation classifier subsequently fine-tuned by the classification loss $L_c$ (i.e., $\widetilde{\mathcal{C}}_m \rightarrow \mathcal{C}_m$). Meanwhile, the class-agnostic aligner is trained through the contrastive loss $L_{align}$. For the second step, the encoder and generator are jointly optimized by $L_c$ and $L_{align}$. Gray indicates frozen modules in each step.

**Generator & Generated Classifier.** Unlike previous FSRE models, our classifier used to discriminate relation classes is not continually optimized across the episodes during training or testing, but is freshly produced for each episode. Concretely, we use a Multi-Layer Perceptron (MLP) as the network generator, which serves as a hypernetwork that aims to produce an initialized N-way relation classifier for each episode. In this way, we can explicitly decouple the upper layers (classifiers) and lower layers (encoder) of our model. Through the two-step training strategy that will be elaborated in Section 3.3, the entire encoder of our model, not just its lower layers, is encouraged to focus on learning general relation knowledge not specific to certain relations in the training set. Specifically, in each episode with N relation classes, the generator takes as input the representation $\boldsymbol{R} = [\boldsymbol{r}_1, \cdots, \boldsymbol{r}_N]$ of all relation descriptions to generate the initializing parameter of a customized relation classifier. The generated classifier is a single linear layer used to predict the relation class of instances in the episode. Notably, unlike previous hypernetwork-based methods (Ha et al., 2016) where the generated module is directly used to make prediction, our produced classifiers are further fine-tuned before actually being used.

### 3.3 Two-step Training

Based on the above model architecture, we expect the generated classifier to acquire relation-specific

knowledge while the encoder and generator can learn more general relation knowledge. To this end, we separate the training of our model into two steps: the fine-tuning of the generated classifiers and the updating of the encoder and generator. Suppose we first sample $M$ episodes from the training set and use the generator to produce an initialized relation classifier for each episode, i.e., $\widetilde{\mathcal{C}}_1, \cdots, \widetilde{\mathcal{C}}_M$.

**Step 1: Classifier Fine-tuning.** Let us denote the generated classifier in the $m$-th episode as $\widetilde{\mathcal{C}}_m$, the classifier is fine-tuned using the corresponding support set (See ② in Figure 4(a)). The training objective of $\widetilde{\mathcal{C}}_m$ is formalized as the following classification loss:

$$
\begin{aligned}
\mathcal{L}_{c_m}(\hat{\boldsymbol{\theta}}_{\widetilde{\mathcal{C}}_m}) &= -\sum_{(x_j, y_j) \in \mathcal{S}_m} y_j \log(\hat{y}_j), \\
\hat{y}_j = \widetilde{\mathcal{C}}_m(x_j) &= \sigma(\boldsymbol{W}_{c_m} \boldsymbol{h}_j + b_{c_m}),
\end{aligned}
\tag{2}
$$

where $\mathcal{S}_m$ refers to the support set of the $m$-th episode, $\hat{\boldsymbol{\theta}}_{\widetilde{\mathcal{C}}_m}$ is the parameter of $\widetilde{\mathcal{C}}_m$ produced by our generator (i.e., $\boldsymbol{W}_{c_m}$ and $b_{c_m}$). In this way, the relation-specific knowledge in the $m$-th episode can be effectively learned by the classifier, resulting in a set of $M$ fine-tuned classifiers $\mathcal{C}_1, \cdots, \mathcal{C}_M$.

**Step 2: Encoder & Generator Updating.** After obtaining all fine-tuned classifiers (i.e., $\mathcal{C}_1, \cdots, \mathcal{C}_M$), we optimize our encoder and gen-

erator using the following objective:

$$\mathcal{L}_c(\boldsymbol{\theta}_\mathcal{E}, \boldsymbol{\theta}_\mathcal{G}) = \frac{1}{M} \sum_m^M \mathcal{L}_{c_m},$$

$$\mathcal{L}_{c_m} = - \sum_{(x_j, y_j) \in \mathcal{Q}_m} y_j \log(\mathcal{C}_m(x_j)), \tag{3}$$

where $\mathcal{Q}_m$ denotes the query set of the $m$-th episode, $\mathcal{L}_{c_m}$ is the classification loss computed by $\mathcal{C}_m$, $\boldsymbol{\theta}_\mathcal{E}$ and $\boldsymbol{\theta}_\mathcal{G}$ represent the parameters of the encoder and generator, respectively. Since all fine-tuned classifiers have been updated at the first step, the parameter deviation disables the gradient directly backpropagated to the generator. To solve this, we adopt a trick that denotes the classifiers in an equivalent form, *i.e.*, $\hat{\boldsymbol{\theta}}_{\mathcal{C}_m}$ of the trained classifier $\mathcal{C}_m$ is denoted as $\hat{\boldsymbol{\theta}}_{\mathcal{C}_m} = \hat{\boldsymbol{\theta}}_{\widetilde{\mathcal{C}}_m} + (\hat{\boldsymbol{\theta}}_{\mathcal{C}_m} - \hat{\boldsymbol{\theta}}_{\widetilde{\mathcal{C}}_m})$. With the help of above loss, the encoder and generator are optimized to enhance the overall classification performance across multiple episodes. By doing so, they are encouraged to learn more general relation knowledge not specific to certain relations.

### 3.4 Class-Agnostic Aligner

Although the above two-step training strategy can already enchance the generalizing capability of our model, there are still only a limited number of episodes involved in the update of the encoder and generator, as in Equation 3. Therefore, the encoder is still possible to overfit the currently sampled $M$ episodes. To avoid this problem, we additionally design a global class-agnostic aligner $\mathcal{A}$ that is an MLP layer. Unlike the classifiers, the parameters of $\mathcal{A}$ are just randomly initialized and then continually optimized across all episodes during the whole training, not freshly produced by the generator in every episode. In this way, the aligner can further encourage the encoder to learn more global knowledge, alleviating its overfitting to the relations within currently sampled $M$ episodes.

Specifically, we first use our encoder $\boldsymbol{E}$ and its exponential moving average (EMA) counterpart $\overline{\boldsymbol{E}}$ to obtain the representations of query and support instances in each of the sampled $M$ episodes. Then, along with the fine-tuning of the generated classifiers, the aligner $\mathcal{A}$ is simultaneously trained using a contrastive loss as follows:

$$\mathcal{L}_{\text{align}}(\boldsymbol{\theta}_\mathcal{A}) = \frac{1}{M} \sum_m^M \mathcal{L}_{a_m},$$

$$\mathcal{L}_{a_m} = \sum_{x_i \in \mathcal{S}_m} \frac{\text{sim}(\mathcal{A}(\boldsymbol{h}_i), \mathcal{A}(\overline{\boldsymbol{h}}_{i'}))}{\sum_{x_j \in \mathcal{Q}_m} \text{sim}(\mathcal{A}(\boldsymbol{h}_i), \mathcal{A}(\overline{\boldsymbol{h}}_j))}, \tag{4}$$

where $\overline{\boldsymbol{h}}$ stands for instance representations obtained from the $\overline{\boldsymbol{E}}$, $\overline{\boldsymbol{h}}_{i'}$ refers to the representation of the positive query instance $x_{i'}$ belonging to the same class as support instance $x_i$. Thus, corresponding to Section 3.3, the overall objective of the first-step training can be written as

$$\mathcal{L}_1 = \Big( \sum_{m=1}^M \mathcal{L}_{c_m}(\boldsymbol{\theta}_{\mathcal{C}_m}) \Big) + \mathcal{L}_{\text{align}}(\boldsymbol{\theta}_\mathcal{A}). \tag{5}$$

For the second-step training, the aligner better encourages the encoder to learn general relation knowledge using a similar contrastive loss:

$$\mathcal{L}_{\text{align}}(\boldsymbol{\theta}_\mathcal{E}) = \frac{1}{M} \sum_m^M \mathcal{L}'_{a_m},$$

$$\mathcal{L}'_{a_m} = \sum_{x_i \in \mathcal{Q}_m} \frac{\text{sim}(\mathcal{A}(\boldsymbol{h}_i), \mathcal{A}(\overline{\boldsymbol{h}}_{i'}))}{\sum_{x_j \in \mathcal{S}_m} \text{sim}(\mathcal{A}(\boldsymbol{h}_i), \mathcal{A}(\overline{\boldsymbol{h}}_j))}. \tag{6}$$

Thereby, the overall objective of the second-step training is formulated as

$$\mathcal{L}_2 = \mathcal{L}_c(\boldsymbol{\theta}_\mathcal{E}; \boldsymbol{\theta}_\mathcal{G}) + \beta \mathcal{L}_{\text{align}}(\boldsymbol{\theta}_\mathcal{E}), \tag{7}$$

where since both loss terms update the encoder, we balance them using a hyperparameter $\beta$.

### 3.5 Inference

During testing, for a new episode from test set, we first use the generator to produce a freshly initialized classifier. Then, the classifier is fine-tuned using the support set to quickly learn the relation-specific knowledge within current episode. Finally, only the encoder and the fine-tuned classifier are used to predict the relation of each query instance.

## 4 Experiment

### 4.1 Datasets

Our model is evaluated on two commonly-used datasets:

- **FewRel 1.0** (Han et al., 2018). It is a large-scale human-annotated FSRE dataset constructed from Wikipedia articles, containing 100 relations. There are 700 instances in each relation. The training, validation and test sets contain 64, 16 and 20 relations, respectively.

- **FewRel 2.0** (Gao et al., 2019c). To evaluate the generalizing capability of our model, we also conduct experiments on FewRel 2.0, whose training set is the same as FewRel 1.0.

Particularly, the test set of FewRel 2.0 is constructed from the biomedical domain (no overlap with relations in th training set) with 25 relations, each of which contains 100 instances.

In each of these two datasets, the training, validation and test sets contain mutually exclusive relation classe sets.

## 4.2 Settings

In previous FSRE studies, there are mainly two types of model settings, differing in whether the model encoder is additionally pre-trained using a noisy Relation extraction(RE) corpus provided by Zhang and Lu (2022). In our experiments, as in previous work, we use BERT (Devlin et al., 2019) to initialize the encoder of our model. For optimization, AdamW (Loshchilov and Hutter, 2019) is used with a linear warmup (Goyal et al., 2017) for the first 10% steps. The learning rates of the encoder and generator are set to 1e-5, while those of the Aligner $\mathcal{A}$ and the generated classifiers are set to 4e-5 and 1e-2, respectively. Following previous work, we set the number of sampled episode $M$ at every training step to 4. It takes about ten hours for the whole training on a single 24 GB NVIDIA RTX 3090 GPU. For testing, we use classification accuracy as the performance metric.

## 4.3 Baselines

We compare our model with the following baseline methods: 1) **Proto-BERT** (Snell et al., 2017), a BERT-based prototypical network model. 2) **MAML** (Finn et al., 2017), a typical meta-learning method. 3) **BERT-PAIR** (Gao et al., 2019c), a similarity-based prediction method, in which each query instance is paired with all support instances. 4) **REGRAB** (Qu et al., 2020), a relation graph-based approach. 5) **TD-Proto** (Yang et al., 2020), a prototypical network model enhanced by entity description. 6) **MTB** (Soares et al., 2019), a BERT-based model further pre-trained using additional *matching the blank* objective. 7) **CP** (Peng et al., 2020), an entity-masked contrastive pre-training framework for FSRE. 8) **HCRP** (Han et al., 2021a), an improved Proto-BERT with a hybrid attention module and a task adaptive focal loss. 9) **SimpleFSRE** (Liu et al., 2022), a prototypical network model enhanced by relation description. 10) **GM_GEN** (Li and Qian, 2022), a graph-based prototypical network model with a similar testing procedure as our model. 11) **LPD** (Zhang and Lu,

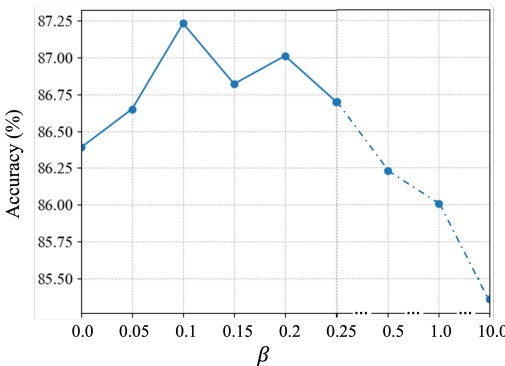

Figure 5: 10-way-1-shot accuracy of our models with different values of $\beta$ on FewRel 1.0 validation set.

2022), a *label prompt dropout* method that effectively exploits the relation description, and filters the pre-training data of CP to conduct more rigorous few-shot evaluation.

## 4.4 Effect of Hyper-parameters $\beta$

The $\beta$ in Equation 7 is an important hyper-parameter, which balances the two loss terms regarding the optimization of our encoder. Thus, we conduct an experiment with different values of $\beta$ on the validation set of FewRel 1.0. From Figure 5, we observe that our model achieves the best performance when $\beta$ is set to 0.1. Hence, we use $\beta = 0.1$ in all subsequent experiments.

## 4.5 Main Results

**Results on FewRel 1.0.** The experimental results on the validation and test sets of FewRel 1.0 are presented in Table 1. In the first part of the table, all the models directly use BERT to initialize their encoder without additional pre-training. For these models, we observe that our model consistently outperforms all other contrast models on FewRel 1.0 dataset, especially surpassing the strongest *GM_GEN* that also fine-tunes a relation classifier for each testing episode. Moreover, the models in the second part also use BERT to initialize their encoder. Notably, they are additionally pre-trained using the noisy RE corpus before the training on Few-shot 1.0. To conduct more rigorous few-shot evaluation, Zhang and Lu (2022) filters the relations in FewRel 1.0 from the noisy RE corpus. Among these models, our model performs significantly better than contrast models in all $N$-way-$K$-shot settings.

In addition, it is noteworthy that our model achieves greater gains on the more challenging 1-shot setting than that on 5-shot setting. All of the above results indicate that our model exhibits

| $N$-way-$K$-shot | 5-way-1-shot | | 5-way-5-shot | | 10-way-1-shot | | 10-way-5-shot | |
|---|---|---|---|---|---|---|---|---|
| Models (*w/o.* pre-train) | val | test | val | test | val | test | val | test |
| Proto-BERT (Snell et al., 2017) | 82.92 | 80.68 | 91.32 | 89.60 | 73.24 | 71.48 | 83.68 | 82.89 |
| MAML (Finn et al., 2017) | 82.93 | 89.70 | 86.21 | 93.55 | 73.20 | 83.17 | 76.06 | 88.51 |
| BERT-PAIR (Gao et al., 2019c) | 85.66 | 88.32 | 89.48 | 93.22 | 76.84 | 80.63 | 81.76 | 87.02 |
| MTB (Soares et al., 2019) | – | 91.10 | – | 95.40 | – | 84.30 | – | 91.80 |
| REGRAB (Qu et al., 2020) | 87.95 | 90.30 | 92.54 | 94.25 | 80.26 | 84.09 | 86.72 | 89.93 |
| TD-Proto (Yang et al., 2020) | – | 84.76 | – | 92.38 | – | 74.32 | – | 85.92 |
| HCRP (Han et al., 2021a) | 90.90 | 93.76 | 93.22 | 95.66 | 84.11 | 89.95 | 87.79 | 92.10 |
| LPD (Zhang and Lu, 2022) | 88.84 | 93.79 | 90.65 | 95.07 | 79.61 | 89.39 | 82.15 | 91.08 |
| SimpleFSRE (Liu et al., 2022) | 91.29 | 94.42 | 94.05 | 96.37 | 86.09 | 90.73 | 89.68 | 93.47 |
| GM_GEN (Li and Qian, 2022) | 92.65 | 94.89 | 95.62 | 96.96 | 86.81 | 91.23 | 91.27 | 94.30 |
| Baseline (our) | 87.04 | 91.03 | 91.82 | 93.87 | 80.07 | 84.98 | 86.95 | 91.13 |
| Ours | 93.35 | 95.21 | 95.94 | 97.19 | 87.41 | 91.59 | 91.71 | 94.54 |
| Models (*w/.* pre-train) | val | test | val | test | val | test | val | test |
| CP[†] (Peng et al., 2020) | 88.29 | 90.85 | 92.77 | 95.60 | 80.50 | 83.89 | 88.61 | 90.61 |
| HCRP-CP[†] (Han et al., 2021a) | 90.89 | 94.23 | 92.90 | 95.77 | 83.17 | 89.69 | 86.43 | 91.94 |
| LPD[†] (Zhang and Lu, 2022) | 93.51 | 95.12 | 94.33 | 95.79 | 87.77 | 90.73 | 89.19 | 92.15 |
| Baseline (our)[*] | 91.13 | 93.71 | 94.36 | 95.93 | 86.11 | 90.31 | 89.55 | 92.49 |
| Ours[*] | **95.46** | **95.92** | **96.59** | **97.48** | **89.34** | **92.01** | **92.46** | **94.90** |

Table 1: Accuracy (%) on the FewRel 1.0 validation / test set. "*w/o.* RE pre-train": the models without additional RE pre-training. "*w/.* RE pre-train": the models whose encoders are additionally pre-trained on the noisy RE corpus (Peng et al., 2020). To conduct more rigorous few-shot evaluation, Zhang and Lu (2022) filters relations contained in FewRE1.0 from the corpus. "†": the results are reported in (Zhang and Lu, 2022).

| Model (*w/o.* pre-train) | 5-way 1-shot | 5-way 5-shot | 10-way 1-shot | 10-way 5-shot |
|---|---|---|---|---|
| Proto-BERT | 40.12 | 51.50 | 26.45 | 36.93 |
| BERT-PAIR | 67.41 | 78.57 | 54.89 | 66.85 |
| HCRP | 76.34 | 83.03 | 63.77 | 72.94 |
| LPD | 77.82 | 86.90 | 66.06 | 78.43 |
| GM_GEN | 76.67 | 91.28 | 64.19 | 84.84 |
| Ours | **78.37** | **91.41** | **66.54** | **84.92** |

Table 2: Accuracy (%) on the FewRel 2.0 test set.

| Model (*w/o.* pre-train) | 5-way 1-shot | 10-way 1-shot |
|---|---|---|
| Ours | **93.35** | **87.41** |
| 1 *w/o.* Generator | 92.06 | 85.82 |
| 2 *w/o.* Two-Step Training | 91.53 | 85.36 |
| 3 *w/o.* Generator & Two-Step | 91.18 | 84.93 |
| 4 *w/o.* Aligner $\mathcal{A}$ | 92.48 | 86.42 |
| 5 *w/o.* EMA | 92.76 | 86.66 |

Table 3: Ablation results on FewRel 1.0 validation set.

higher generalizing capability that can better deal with data scarcity issue in harsher few-shot scenarios.

**Results on FewRel 2.0.** As mentioned in Section 4.1, FewRel 2.0 is a more difficult dataset whose training set and test set not only contain mutually exclusive relation classes but also come from different domains. From Table 2, we observe that our model still consistently outperforms all contrast models. These results further demonstrate the superiority of our model.

### 4.6 Ablation Study

We further conduct extensive ablation studies by removing different components of our model to comprehend their different impacts. We compare our model with the following variants in Table 3.

(1) *w/o. Generator*. In this variant, we remove the network generator and employ a shared classifier across all episodes. As shown in **Line 1**, this leads to a significant performance drop of 1.29 and

1.59 points in 5-way and 10-way settings, respectively. These results indicates that the generalizing capability of a shared classifier is limited, demonstrating the effectiveness of our network generator. Particularly, our generator can generate a suitable initialized relation classifier for each episode. Subsequently, the fine-tuning of these classifiers encourages the model to effectively learn relation-specific knowledge from only a handful of support instances, thus quickly adapting to unseen relations.

(2) *w/o. Two-Step Training*. During training, we separate the training of our model model into two steps: the fine-tuning of the generated classifiers and the update of the encoder and generator. To verify the effectiveness of this strategy, in this variant, we turn to simultaneously optimize the classifier, encoder and generator in each episode. As illustrated in **Line 2**, this variant causes a significant performance decline. This suggests that simultaneously optimizing the classifier, encoder,

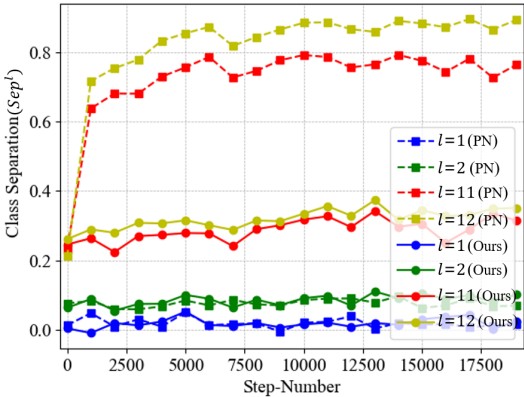

Figure 6: The comparison between conventional Prototypical Network (PN) model and ours in terms of class separation $Sep^l$ at lower and upper layers.

and generator leads to the encoder and generator learning more relation-specific knowledge, thus over-adapting to the relations in the training set.

(3) *w/o. Generator & Two-Step*. When we remove the generator from our model and simultaneously optimize the whole model in each episode, the performance drops by 2.17 and 2.48 points in 5-way and 10-way settings, respectively (See **Line 3**). These results suggest that our generator and two-step training strategy both crucially contribute to the generalizing ability of our model.

(4) *w/o. Aligner $\mathcal{A}$*. The class-agnostic aligner $\mathcal{A}$ aims to provide our model with more global general relation knowledge across the whole training process. To verify its effectiveness, we remove it from our model and the performance also decreases (See **Line 4**). This suggests that the aligner $\mathcal{A}$ can indeed enhance model generalization.

(5) *w/o. EMA*. It is noteworthy that the EMA encoder $\overline{E}$ integrates the historical relation knowledge from previously learned episodes during the whole training. In this variant, the EMA operation is removed and the performance becomes inferior to *Ours* (See **Line 5**). It indicates that, with the help of $\overline{E}$, our aligner $\mathcal{A}$ is able to provide our encoder $E$ with more global general relation knowledge in a more efficient manner.

### 4.7 Comparison of Class Separation with Our Model

To further verify the effectiveness our proposed HyperNetwork-based decoupling approach, we make comparison of the layer-wise class separation $Sep^l$ between the encoder of a conventional prototypical network (PN) FSRE model and that of ours. For clarity, we only depict the degrees of

class separation in bottom-2 and top-2 layers (Layers 1, 2 ,11 and 12) in Figure 6. From the figure, we can observe that, in lower layers (*i.e.*, Layers 1 and 1), the degree of class separation shows little difference between the PN model and ours. However, in upper layers (*i.e.*, Layers 11 and 12), the class separation of our model is consistently much lower and vary moderately during the whole training process. This indicates that our entire encoder, not just its lower layers, focuses on learning more general relation knowledge, thus exhibiting less bias towards relation classes in the training set. For the details about all the layer-wise class separation in our model, please refer to Appendix A.

## 5 Related Work

Relation extraction(RE) is a critical and fundamental task in natural language processing (NLP), which aims to identify the semantic relations between two entities within a given text (Xue et al., 2019; Han et al., 2020; Chen et al., 2022; Zhang et al., 2022, 2023b,a). However, conventional approaches are highly dependent on a large amount of labeled data and cannot deal with unseen relation classes well. Therefore, recent studies (Han et al., 2018; Gao et al., 2019c) turn to Few-Shot Relation Extraction (FSRE) that aims to train a model to classify instances into novel relations with only a handful of training examples.

Most existing studies (Gao et al., 2019a; Yang et al., 2020; Zhang and Lu, 2022) employ Prototype Networks for FSRE, which aims to learn a suitable prototypical vector for each relation using a handful of annotated instances. Gao et al. (2019b) employs an attention mechanism to enhance the robustness of the prototype network to noisy data. Qu et al. (2020) proposes a Bayesian meta-learning method with an external global relation graph to model the posterior distribution of relational prototypes. Han et al. (2021b) focuses on enhancing the performance of Prototype Network on complex relations through an adaptive focal loss and a hybrid network. Moreover, some studies (Yang et al., 2020; Wang et al., 2020; Han et al., 2021b) use supplementary information about entities and relations, such as relation descriptions, to enhance the prototype vectors of relations. Despite impressive results achieved, these methods still tends to overfit relation classes appearing in the training set, which limits their generalizing capability to new relations. In this paper, inspired by (He et al., 2020; Yin

et al., 2022), we propose a HyperNetwork-based decoupling method along with a two-step training strategy to prevent overfitting of the FSRE model to the relations within the training set.

On the other hand, some studies (Soares et al., 2019; Peng et al., 2020; Dong et al., 2021; Wang et al., 2022) focus on further training pre-trained language models (PLMs) using noisy RE datasets. Soares et al. (2019) collect a large-scale pre-training dataset and propose a *matching the blanks* pre-training paradigm. Peng et al. (2020) proposes an entity-masked contrastive pre-training framework for FSRE. Wang et al. (2022) introduces three structure pre-training tasks to pre-train the large language model (GLM with 10B parameters), allowing it to better comprehend structured information in text. Unlike several other studies, Zhang and Lu (2022) introduces a more rigorous few-shot evaluation scenario by filtering relations contained in FewRE 1.0 from the pre-trained corpus. Meanwhile, they propose a label prompt dropout method to prevent the model from overfitting to the relation description. These methods are compatible with our model, which can provide our model with a better pre-trained encoder.

## 6 Conclusion

In this paper, we propose a HyperNetwork-based Decoupling approach to improve the generalizing capability of FSRE models. Specifically, our model consists of an encoder, a network generator (for generating relation classifiers) and the generated classifiers. Our generator aims to generate a properly initialized relation classifier for each episode, allowing our model can quickly adapt to new relations. Meanwhile, we design a two-step training strategy along with a class-agnostic aligner, in which the generated classifiers focus on acquiring relation-specific knowledge while the encoder is encouraged to learn more general relation knowledge. In this way, the roles of upper and lower layers in an FSRE model are explicitly decoupled, thus enhancing its generalizing capability during testing. Experiments on two public FSRE datasets and extensive ablation studies show that our model consistently outperforms all competitive baselines.

## Acknowledgments

The authors would like to thank the three anonymous reviewers for their comments on this paper. This research was supported in part by the National Natural Science Foundation of China under Grant Nos. 62076211, U1908216, 62276219, and 61573294.

## Limitations

The limitations of our method mainly include following two aspects: 1) Our method is only examined on the FSRE task, while whether it is able to generalize to other tasks, such as intent classification and image classification, is not yet explored in this paper. 2) We did not consider non-of-the-above scenarios where a query instance may not belong to any class in the support set.

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

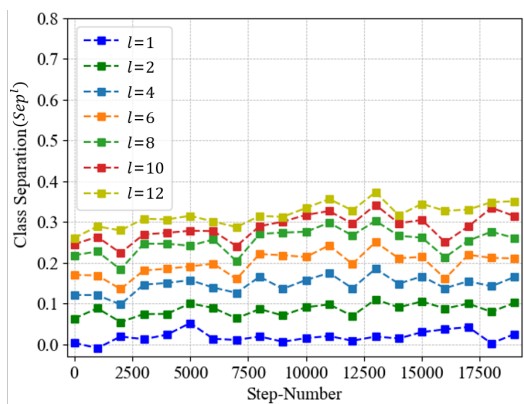

Figure 7: Layer-wise class separation $Sep^l$ in our 12-layer encoder $E$ with respect to the training steps.

# Appendix

## A   Layer-wise Class Separation of our model

In this section, we investigate the relation classes separation within our encoder in detail, as we done in the preliminary study section. As illustrated in Figure 7, the degree of class separation $Sep^l$ in each layer of our encoder remains stable throughout the training process. In particular, at 12-th layer, our encoder has significantly lower the class separation than the prototype network. This intuitively demonstrates that our training strategy can effectively train our entire encoder, not just its lower layers, to learn more general relation knowledge.