# OpenReview forum: "HyperNetwork-based Decoupling  to Improve Model Generalization for Few-Shot Relation Extraction"
_EMNLP/2023/Conference — EMNLP 2023 Main_

### Official Review · Reviewer_ePtg · 2023-08-04

**Typos Grammar Style And Presentation Improvements:** 173
**Soundness:** 4

**Excitement:**

4: Strong: This paper deepens the understanding of some phenomenon or lowers the barriers to an existing research direction.

**Missing References:**

The authors should cite MoCo since they are using contrastive learning + EMA:

He et al. Momentum Contrast for Unsupervised Visual Representation Learning



**Paper Topic And Main Contributions:**

This paper tackles the few-shot relation extraction (FSRE) problem (to be more specific, FewRel and FewRel 2.0 datasets). The task setting is that during test time, an episode of N-class K-shot data will be sampled and the model will be tested on the (previously unseen) N tasks.

The authors first conduct preliminary studies to show that in FSRE models, the representations at early layers are usually quite similar across classes but are more distinct at last layers. Then they propose to use a hypernetwork (an MLP) to generate the class embeddings (for the final linear classification), in contrast to conventional approaches like Prototypical Network (averaging class instances to get the class embedding). There are three key components of the proposed method: (1) the generator which takes the encoded relation description as input and outputs the relation embedding; (2) two-step training where they first fix the encoder and only fine-tune the relation embedding, and then then fix the embedding to fine-tune the generator and encoder; (3) an "aligner" which is used for a momentum-encoder-style contrastive learning.

The authors conduct extensive experiments on FewRel and FewRel 2.0 and demonstrate very strong results compared to a number of baselines. Even compared to models using RE pre-training, the proposed method is still very effective. There is also a comprehensive ablation study to demonstrate the effectiveness of each component.


**Questions For The Authors:**

(1) How would the aligner itself perform if used as the few-shot classifier? I would be interested to see how well contrastive learning can do on few-shot classification tasks.

(2) For the two-step optimization: it makes sense to do the theta_cm^ + (theta_cm - theta_cm^) trick (I imagine the second term has stop gradient), but if the embedding fine-tuning was already very good, doesn't this stop optimizing the generator at all (because theta_cm - theta_cm^ already did all the heavy lifting)? Would something like using gradient of the gradient make more sense (but of course it's way more complicated)?

(3) I didn't find the details for how to fine-tune the classifier embedding (sorry if I missed it). Can you elaborate?

**Reasons To Accept:**

(1) The proposed method is based on a sold preliminary observation, is well motivated, and is intuitive.

(2) The experiment is very comprehensive and the results are very strong, especially that this method can still bring improvement over RE-pre-trained models.

(3) The ablation study shows that every component is essential to the success of the method.

**Reasons To Reject:**

(1) This is nitpicking, but it would be interesting to see how large language models would perform on such a task. Also, even with a small model like BERT, having a prompt would probably make the performance better.


**Reproducibility:**

5: Could easily reproduce the results.

**Reviewer Confidence:**

4: Quite sure. I tried to check the important points carefully. It's unlikely, though conceivable, that I missed something that should affect my ratings.

---

> ### Author Rebuttal · Authors · 2023-08-29
>
> > $\textbf{Q1}$: I would be interested to see how well the aligner and large language models perform on FSRE tasks.
>
> We report the performances of the aligner and ChatGPT as follows:
>
> |   |    |    |    |    |
> | :--- | :----: | :----: | :----: | :----: |
> | Model | 5-way-1-shot | 5-way-5-shot | 10-way-1-shot | 10-way-5-shot |
> |  |val / test | val / test | val / test | val / test|
> | Aligner | 90.12 / 93.31 | 92.93 / 95.33 | 83.87 / 88.72 | 87.55 / 91.94 |
> | ChatGPT | 81.33 / 83.01 | 83.97 / 86.56 | 71.22 / 75.19 | 74.71 / 78.11 |
> | Ours | 93.35 / 95.21 | 95.94 / 97.19 | 87.41 / 91.59 | 91.71 / 94.54 |
> |   |    |    |    |    |
>
> where Ours denotes the performance of our model as reported in our paper.
>
> We evaluate ChatGPT on 1,000 episodes randomly sampled from the FewRel 1.0 validation and test sets. Meanwhile, we investigate four commonly-used prompt templates and report the performance of the optimal template.
>
> As shown in the table above, our model outperforms the aligner and ChatGPT, which further demonstrates the superiority of our method.
>
>
> > $\textbf{Q2}$: For the two-step optimization: it makes sense to do the $\boldsymbol{\hat{\theta}}\_{\mathcal{C}\_m} = \boldsymbol{\hat{\theta}}\_{\mathcal{\widetilde{C}}\_m}+(\boldsymbol{\hat{\theta}}\_{\mathcal{C}\_m} - \boldsymbol{\hat{\theta}}\_{\mathcal{\widetilde{C}}\_m})$ trick, but if the embedding fine-tuning was already very good, doesn't this stop optimizing the generator at all? Would something like using gradient of the gradient make more sense?
>
> At each model optimization, our generator first generates a new initial classifier for each episode, and then the generated classifier is fine-tuned on the support set of its corresponding episode. Finally, the generator is optimized using the loss of the fine-tuned classifier on the query set. Since the classifier is regenerated each time and fine-tuned on the support set (rather than the query set), this loss for training the generator is usually non-zero. Therefore, our generator can be optimized continuously.
>
> Moreover, we believe that using gradient of the gradient is an interesting idea. In future work, we will attempt to employ this approach.
>
> > $\textbf{Q3}$: I didn't find the details for how to fine-tune the classifier embedding (sorry if I missed it). Can you elaborate?
>
> The fine-tuning process of the generated classifier is described in detail on Lines 277 to 289 of our paper. Specifically, for each episode, we first use the encoder to obtain the representations of the instances and relation descriptions in the episode. Based on these representations of relation descriptions, we then employ the generator to produce an initial relation classifier for the episode. Finally, we fine-tune the classifier by utilizing its loss on the episode's support set (See Equation 3 in our paper).
>
>
> > $\textbf{Q4}$: Typos and Missing References
>
> Thanks. We will fix them.

---

### Official Review · Reviewer_bDys · 2023-08-04

**Soundness:** 4

**Excitement:**

4: Strong: This paper deepens the understanding of some phenomenon or lowers the barriers to an existing research direction.

**Missing References:**

Yin, Li, Juan M. Perez-Rua, and Kevin J. Liang. "Sylph: A hypernetwork framework for incremental few-shot object detection." Proceedings of the IEEE/CVF Conference on Computer Vision and Pattern Recognition. 2022.

**Paper Topic And Main Contributions:**

Most existing approaches to few-shot relation extraction(FSRE) employ Prototypical Networks, which usually overfit the relation classes in the training set and perform poorly on unseen relations. This paper explores what restricts the model's generalizing capability and proposes a HyperNetwork-based Decoupling approach for FSRE. The model consists of three components: an encoder for obtaining the representation of relations and instances, a network generator for generating the initialized relation classifiers, and the generated classifiers to predict the relation of each query instance. With a two-step training strategy and a class-agnostic aligner, the proposed model alleviates the overfitting issue and exhibits higher generalization. Experimental results and ablation studies demonstrate the effectiveness of the method.

**Questions For The Authors:**

Please see "Reasons to Reject" above.

**Reasons To Accept:**

1.  the motivation of this paper is clear, and the proposed method seems novel to me.
2.  this paper is easy to follow
3.  extensive experiments with promising results

**Reasons To Reject:**

1.  The contributions of this work are suggested to be explicitly highlighted in the introduction. The audiences need to take time to dig out them.

2. The differences between the proposed work and the below work should be discussed.

Yin, Li, Juan M. Perez-Rua, and Kevin J. Liang. "Sylph: A hypernetwork framework for incremental few-shot object detection." Proceedings of the IEEE/CVF Conference on Computer Vision and Pattern Recognition. 2022.

**Reproducibility:**

4: Could mostly reproduce the results, but there may be some variation because of sample variance or minor variations in their interpretation of the protocol or method.

**Reviewer Confidence:**

3: Pretty sure, but there's a chance I missed something. Although I have a good feel for this area in general, I did not carefully check the paper's details, e.g., the math, experimental design, or novelty.

**Typos Grammar Style And Presentation Improvements:**

Line 595: "the roles of upper and lower layers in an FSRE model is explicitly decoupled". Here "is" should be "are".

---

> ### Author Rebuttal · Authors · 2023-08-29
>
> > $\textbf{Q1}$: The contributions of this work are suggested to be explicitly highlighted in the introduction. The audiences need to take time to dig out them.
>
> Thank you very much for your valuable suggestion. In the revised version of our paper, we will include our contributions within the introduction section.
>
> > $\textbf{Q2}$: The differences between the proposed approach and “Sylph: A hypernetwork framework for incremental few-shot object detection” should be discussed.
>
> Thank you for your reminder. In the revised version of our paper, we will provide a comparison between *Sylph* and our approach.
>
> The differences between *Sylph* and our approach can be summarized as follows:
> 1. *Sylph* focuses on the incremental few-shot object detection, where the model is required to adapt to novel classes without forgetting previously seen classes. Instead, our approach aims to enhance the model's generalization ability to novel relations in the test set by preventing the model from overfitting the relations (previously seen classes) in the training set.
> 2. Although both *Sylph* and our approach employ hypernetwork, the structures of the two hypernetworks are notably distinct, and the modules generated by them also exhibit significant differences. *Sylph* employs a multi-layer convolutional neural network as a hypernetwork, and the classifiers generated by this hypernetwork are non-trainable. We apply a simple multi-layer perceptron as a hypernetwork, and the generated classifiers are further fine-tuned before actually being used. This trainable classifier enables our model to quickly adapt to new relations during testing.
> 3. Our training strategy differs from that adopted by *Sylph*. *Sylph* first pre-trains the image encoder, and then freezes the encoder and train its hypernetwork to produce classifiers. Instead, we propose a two-step training strategy to train our model, where the classifiers and the encoder are alternately trained in a decoupling manner. As demonstrated in Figure 5 and Line 2 of Table 3, our training strategy can effectively prevents the encoder from overfitting the relations in the training set, enabling our model to better generalize to new relations in the test set.
>
> >  $\textbf{Q3}$: Typos and Missing References
>
> Thanks. We will fix them.

---

### Official Review · Reviewer_TfK4 · 2023-08-18

**Soundness:** 3

**Excitement:**

3: Ambivalent: It has merits (e.g., it reports state-of-the-art results, the idea is nice), but there are key weaknesses (e.g., it describes incremental work), and it can significantly benefit from another round of revision. However, I won't object to accepting it if my co-reviewers champion it.

**Missing References:**

https://aclanthology.org/2022.findings-acl.67v2.pdf

**Paper Topic And Main Contributions:**

This paper proposed a hypernetwork-based decoupling approach for few-shot relation extraction. This approach has demonstrated effectiveness in FewRel 1.0 and 2.0 datasets.

**Questions For The Authors:**

1. Given that the experimental improvements are not that significant, how many times did you run the experiments?
2. Please conduct statistical significant test with multiple experiment runs
3. deepstruct is a popular and open-sourced paper. Why deepstruct was not cited or compared?

**Reasons To Accept:**

This paper adopts a relatively new approach in the context of few-shot relation extraction.

**Reasons To Reject:**

This paper adopts existing components in FSRE, however, the novelty of applying previously proposed techniques is limited. The experimental improvements of the proposed approach are also marginal, especially when comparing against GM_GEN. Besides, this paper did not cite the existing SOTA deepstruct, which also adopts relation pre-training. I think it is necessary to justify the reason for not citing this paper.

**Reproducibility:**

3: Could reproduce the results with some difficulty. The settings of parameters are underspecified or subjectively determined; the training/evaluation data are not widely available.

**Reviewer Confidence:**

3: Pretty sure, but there's a chance I missed something. Although I have a good feel for this area in general, I did not carefully check the paper's details, e.g., the math, experimental design, or novelty.

---

> ### Author Rebuttal · Authors · 2023-08-29
>
> > $\textbf{Q1}$ : This paper adopts existing components in FSRE, however, the novelty of applying previously proposed techniques is limited.
>
> The contributions of this paper are summarized as follows:
> 1. We propose a hypernetwork-based decoupling approach for FSRE, which uses a hypernetwork (the network generator) to explicitly decouple the lower layer (the encoder) and upper layer (the generated relation classifiers) of the model. As illustrated in Line 1 of Table 3, this decoupling model architecture can effectively enhance the generalization of our model to novel relations. **It is worth noting that** this hypernetwork-based decoupling architecture has not been explored in existing FSRE studies. Moreover, unlike previous hypernetwork-based methods where the generated module are usually not trainable, our generated classifiers are further fine-tuned before actually being used, which enables our model to quickly adapt to new relations.
>
> 2. We propose a two-step training strategy along with a class-agnostic aligner, in which the generated classifiers focus on acquiring relation-specific knowledge while the encoder is encouraged to learn more general relation knowledge. In this way, we can effectively prevent the encoder from overfitting the relations in the training set, enabling our model to better generalize to new relations in the test set. This is effectively demonstrated in Figure 5 and Line 2 of Table 3. **As far as we know**, this training strategy is the first attempt in FSRE to alternately train the encoder and classifiers in a decoupling manner.
>
>
>
> > $\textbf{Q2}$ : Compared with *GM_GEN*, the improvements of the proposed method are marginal. Please conduct statistical significant test and multiple experiment runs.
>
> According to the open-source code from *GM_GEN*, we find that *GM_GEN* employs distinct distance functions in different N-way-K-shot settings. To ensure a fair comparison, we adopt the same distance function as *GM_GEN* and re-report the performance of our model as follows:
>
> |   |    |    |    |    |
> | :--- | :----: | :----: | :----: | :----: |
> | Model | 5-way-1-shot | 5-way-5-shot | 10-way-1-shot | 10-way-5-shot |
> |  |val / test | val / test | val / test | val / test|
> | Baseline (our)| 87.04 / 91.03 | 91.82 / 93.87 | 80.07 / 84.98 | 86.95 / 91.13 |
> |   |    |    |    |    |
> | GM_GEN | 92.65 / 94.89 | 95.62 / 96.96 | 86.81 / 91.23 | 91.27 / 94.30 |
> | Ours-old | 93.35 / 95.21 | 95.94 / 97.19 | 87.41 / 91.59 | 91.71 / 94.54 |
> | ∆ | +0.70 / +0.32 | +0.32 / +0.23 | +0.60 / +0.36 | +0.44 / +0.24 |
> |   |    |    |    |    |
> | GM_GEN | 92.65 / 94.89 | 95.62 / 96.96 | 86.81 / 91.23 | 91.27 / 94.30 |
> | Ours-new | 93.75±0.14 / 95.53±0.05 | 96.86±0.21 / 97.55±0.10 | 87.80±0.19 / 91.78±0.07 | 92.50±0.27 / 94.89±0.13 |
> | ∆ | +1.10 / +0.64 | +1.24 / +0.59 | +0.99 / +0.55 | +1.23 /+0.59 |
> | $p$ value | 0.0036 / $-$ | 0.0027 / $-$ | 0.0041 / $-$ | 0.0023 / $-$ |
> |   |    |    |    |    |
>
> We run experiments 5 times with different random seeds and report the mean and standard deviation on the FewRel 1.0 validation and test sets. Meanwhile, we perform statistical significance tests and report the corresponding $p$-values.
>
> As shown in the table above, using the same distance function, our model significantly outperforms *GM_GEN* ($p$<0.01) by a larger margin (See ∆).
> Moreover, given the apparent discrepancy between our model and *GM_GEN*, we believe that the performance improvement of our model over our Baseline is more reliable indicator of the efficacy of our approach.
>
>
> > $\textbf{Q3}$ : Why *Deepstruct* was not cited or compared?
>
> We appreciate your reminder. In the revised version of our paper, we will cite this work.
>
> Next, we provide a detailed comparison between *Deepstruct* and our approach：
> 1. The primary focus of *Deepstruct* and our approach are notably dissimilar. Similar to *CP* and *LPD*, *Deepstruct* focuses on further training pre-trained language models (PLMs) to improve its performance on structured prediction tasks. In contrast, our approach aims to prevent the PLM-based encoder from overfitting the relations in the training set, thus enabling our model to better generalize to new relations in the test set. Moreover, our approach usually is compatible with such methods that further train PLMs. As shown in the bottom cell of Table 1, our method can effectively enhance the performance of *LPD*.
> 2. *Deepstruct* constructed its pre-training dataset in a similar manner to the FewRel 1.0 dataset, and did not exclude relations present in the FewRel 1.0 test set from its pre-training data. *LPD* has verified that the performance of the model will be overestimated under this setting. Instead, we follow *LPD* to conduct a more rigorous few-shot evaluation by excluding relations in the FewRE1.0 test set from the pre-training data. Consequently, it would be unfair to conduct a direct comparison between the performance of *Deepstruct* and that of our approach.
> 3. *Deepstruct* focuses on enhancing the ability of large PLM (10B GLM) to understand structural information within text, while our method aims to enhance the generalization of small PLM (BERT-base) to new relations. The large gap in the number of parameters between *Deepstruct* and our approach makes a direct comparison between the two unfair.

---

### Meta-Review · Area_Chair_fLSq · 2023-09-19

**Recommendation:** 4

**Metareview:**

This research paper introduces a novel approach to tackle few-shot relation extraction using a hypernetwork-based decoupling method. The model comprises three key components: an encoder, responsible for extracting representations of relations and instances; a network generator, which produces the initial relation classifiers; and the generated classifiers themselves, used to predict relations for each query instance. By employing a two-step training strategy and a class-agnostic aligner, the proposed model effectively addresses the problem of overfitting and demonstrates enhanced generalization capabilities.

---

### Decision · Program_Chairs · 2023-10-07

**Decision:**

Accept-Main

**Comment:**

This research paper introduces a novel approach to tackle few-shot relation extraction using a hypernetwork-based decoupling method. The model comprises three key components: an encoder, responsible for extracting representations of relations and instances; a network generator, which produces the initial relation classifiers; and the generated classifiers themselves, used to predict relations for each query instance. By employing a two-step training strategy and a class-agnostic aligner, the proposed model effectively addresses the problem of overfitting and demonstrates enhanced generalization capabilities.